# A New Impregnated Adsorbent for Noble Metal Ion Sorption

**DOI:** 10.3390/molecules28166040

**Published:** 2023-08-13

**Authors:** Zbigniew Hubicki, Karolina Zinkowska, Grzegorz Wójcik

**Affiliations:** Department of Inorganic Chemistry, Institute of Chemical Sciences, Faculty of Chemistry, Maria Curie-Skłodowska University, Maria Curie-Skłodowska Sq. 2, 20-031 Lublin, Poland; 283832@office.umcs.pl

**Keywords:** noble metals, Aliquat 336, Nitrolite, green adsorbent, warm impregnation

## Abstract

Noble metals (NM) such as gold, platinum, palladium, and rhodium are widely applied in the electronics and automotive industries. Thus, the search for cheap and selective sorbents for noble metals is economically justified. Nitrolite does not sorb noble metal ions. A new impregnated sorbent was prepared. The natural sorbent Nitrolite was impregnated with Aliquat 336 using a new warm impregnation method. After the impregnation process, Nitrolite adsorbed platinum(IV), palladium(II), and gold(III) ions from the chloride solutions. The values of the sorption capacity for palladium(II) and platinum(IV) ions were 47.63 mg/g and 51.39 mg/g, respectively, from the 0.1 M HCl model solution. The sorption capacity for gold(III) ions was estimated to be 73.43 mg/g from the 0.1 M HCl model solution. An exhausted catalytic converter was leached, and platinum(IV), palladium(II), and rhodium(III) were transferred to the chloride solution. The impregnated sorbent Nitrolite–Aliquat 336 was used in the investigations of the platinum(IV), palladium(II), and rhodium(III) ions’ sorption from a real solution. The impregnated sorbent Nitrolite–Aliquat 336 proved to be suitable for the recovery of platinum(IV) and palladium(II) ions, whereas rhodium ions were not sorbed from the leached solutions. Notably, 1 M thiourea in the 1 M HCl solution desorbed platinum(IV), palladium(II), and gold(III) above 94%.

## 1. Introduction

Noble metals such as platinum, palladium, rhodium, and gold are thermally and chemically resistant. Owing to their unique properties, they are used for the production of automotive exhaust gas converters [1]. Enantioselective domino reactions are promoted by the chiral catalysts derived from noble metals [2]. Precious metals are also a capital investment. During the ongoing war in Eastern Europe, the prices of precious metals are very high [3]. There are many ways to isolate precious metals through the use of a leaching solution. Platinum can be removed via the precipitation of ammonium hexachloroplatinate [4]. Similar precipitation has been used for platinum removal from a chloride solution [5]. Noble metals have a positive redox potential that can be reduced by negative redox potential metals. The cementation process can be used for the removal of noble metals from solutions. Gold thiocyanate complexes were removed from a thiocyanate solution via cementation with iron powder [6]. Gold can be reduced by zinc or organic acids such as oxalic acid. Aside from precipitation, solvent extraction is used to separate precious metals. Cyanex 272 (bis(2,4,4-trimethylpentyl)phosphinic acid) was used for the separation of gold(III) ions from platinum(IV), palladium(II), rhodium(III), and iridium(III) in hydrochloric acid solutions (0.5–9 M) [7]. Palladium(II) ions were separated from platinum(II), rhodium(III), and iridium(III) in a concentrated 6M hydrochloric acid solution using LIX63 (5,8-diethyl-7-hydroxydodecane-6-oxime) [8]. TBP (tri-n-butylphosphate) and TOPO (tri-n-octylphosphine oxide) were tested for the separation of platinum(IV) from rhodium(III) ions in hydrochloric acid solutions. From an economical point of view, considering the separation process, this indicates that TBP is better than TOPO for the separation of Pt and Rh from chloride solutions [9]. The extractants that contain sulfur-donor atoms can separate noble metals. Cyanex 471X (tri-isobutylphosphine sulfide) is able to separate platinum(IV) ions from rhodium(III) in hydrochloric acid solutions [10]. In addition to extractants containing S- and P-donor atoms, amines can also be used for the sorption and separation of noble metals. Pd(II) and Pt(IV) have been extracted from hydrochloric solutions, but both pure palladium and platinum can be obtained using selective stripping methods [11]. The other method used for the separation of precious metals is the ion exchange method. Ion exchangers are composed of organic skeletons and functional groups. Functional groups most often contain N-, S-, or P-donor atoms and are capable of noble metal ion sorption. Diaion WA21J is a commercial polyamine-type anion exchanger containing N-donor atoms. This anion exchanger is capable of gold(III) ion sorption from the leaching solution (aqua regia). An elution of gold(III) ions from WA21J was prepared using a diluted thiourea solution [12]. A chelating-type resin with thiouronium functional groups (XUS 43,600.00) was used for the selective removal of platinum(IV), palladium(II), and rhodium(III) ions from the process’s leaching solution. XUS 43,600.00 was found to show a better adsorption action for platinum(IV) and palladium(II) chloride complexes than for rhodium(III) chloride complexes [13]. Anion exchangers and chelating-type resins are effective for noble metals sorption; however, they are costly. There is a group of impregnated sorbents that combine the benefits of liquid-liquid extraction and ion-exchange sorbents. Solvent-impregnated resins (SIRs) enable the incorporation of the extractant into the resin matrix. SIRs are characterized by sorption properties; therefore, they are used as adsorbents. SIRs allow for the incorporation of well-known or new extractants into the polymer resin; they circumvent the problem of creating the third phase in the extraction, which is the acquisition of new sorption properties. Moreover, they can be prepared using simple methods [14]. Solvent-impregnated resins (SIRs) are one of the most exciting research areas due to their huge potential for applications in processes such as the removal of dyes and phenolic compounds from water, as well as for metal ion sorption [15]. SIRs can be used for gold(III) ion sorption from leaching solutions. Gold(III) ions were recovered effectively from a chloride solution after the digestion of the modular connector RJ 45 using Purolite MN 202 after the (Cyanex 272) impregnation process [16]. During the impregnation process, Cyanex 272 was diluted in acetone. Hexane [17] and chloroform [18] were also used as extractant solvents. One of the objectives of this paper was to develop a method of impregnation without the use of potentially toxic organic solvents, e.g., chloroform. Aliquat 336 was used for the impregnation process. Due to its low volatility, it can be considered a more environmentally friendly and safer alternative to organic solvents [19]. As the matrix for the impregnation process, Nitrolite sorbent was used. Nitrolite is a natural sorbent enabling the sorption of Cr(III), La(III), Ce(III), Pr(III), and Nd(III) ions [20]. The other goal was the use of Nitrolite impregnated by Aliquat 336 for the removal of noble metal ions from chloride solutions.

## 2. Results and Discussion

### 2.1. Preparation of the Impregnated Sorbent Nitrolite–Aliquat 336

The impregnated sorbent Nitrolite–Aliquat 336 was prepared using a new method. Aliquat 336 was not diluted in an organic solvent but was heated and mixed with Nitrolite. This method provides a simple way to change the properties of the natural sorbent Nitrolite. It can be called warm impregnation, which is described in the Methods section.

#### Analysis of the Impregnated Sorbent Nitrolite–Aliquat 336

After the impregnation process, the concentration of Aliquat 336 in Nitrolite was determined via conductivity titration. The concentration of Aliquat 336 in Nitrolite was 0.25 g/g. A 0.25 g concentration of Aliquat 336/g Nitrolite is a high value. During the solvent impregnation of Purolite MN202 using Cyanex 272, 0.105 g of Cyanex272/g Purolite MN202 was found [16]. This comparison demonstrates the effectiveness of the new impregnation method. Aside from titration, FTIR-ATR spectra were recorded to confirm the presence of Aliquat 336 in Nitrolite.

The FTIR spectra of Aliquat 336, Nitrolite, and Nitrolite impregnated with Aliquat 336 are presented in Figure 1. All three spectra were compared with each other. The symmetric and asymmetric stretching vibrations of C-H occur at 2924 cm^−1^ and 2855 cm^−1^, which is evidenced in Aliquat 336 spectrum [21]. The same bands were observed in the Nitrolite–Aliquat 336 FTIR spectrum. The band at 1624 cm^−1^ is due to the stretching vibrations of the hydroxyl groups from H_2_O contained in the Nitrolite–Aliquat 336 sorbent. In the spectra of pure Aliquat 336 and Aliquat 336-impregnated Nitrolite, a characteristic peak for quaternary ammonium salts was observed (1465 cm^−1^) [22]. Asymmetric stretching vibrations of SiO_4_ and AlO_4_, which were found in Nitrolite, corresponded to a narrow peak at 1015 cm^−1^. The rocking vibrations of (-CH_2_-)_n_ from Aliquat 336 were detected at 729 cm^−1^. The band at 594 cm^−1^ is attributed to the pseudo-lattice vibrations of Nitrolite [23]. The stretching vibrations of O-H might be caused by the incidence of a wide band at 3396 cm^−1^.

The SEM images of Nitrolite are presented in Figure 2a,b. As can be seen, Nitrolite is a porous material. The SEM images of Nitrolite–Aliquat 336 are presented in Figure 2c,d. The surface of the impregnated sorbent is smooth, and it can be seen that pores are filled with Aliquat 336. The SEM results in Figure 2 confirm that Aliquat 336 was present on the Nitrolite surface, and the impregnation process was efficient.

### 2.2. Kinetics of Noble Metal Ion Sorption on the Impregnated Sorbent Nitrolite–Aliquat 336

The kinetics of noble metal ion sorption on the impregnated sorbent Nitrolite–Aliquat 336 was investigated. Noble metal ions’ concentrations were determined using the ICP-OES method. The influence of the removal percentage (R%) on the contact time was determined in the HCl concentration range of 0.1–6 M. The recovery percentage (R%) of gold(III) was calculated as follows:(1)$$R\%=\frac{C}{C_{0}}\times 100\%$$
where *C* is the concentration of the adsorbed noble metal ions calculated from the difference in the solution concentration before and after the sorption process, and *C*_0_ is the initial concentration of noble metal ions. The influence of R% on the phase contact time is presented in Figure 3. The results in Figure 3 were obtained from gold(III), palladium(II), platinum(IV), and rhodium(III) mixtures. As can be seen, rhodium ions were not sorbed by the impregnated sorbent Nitrolite–Aliquat 336. This provides evidence of strong competition among noble metal ions in terms of functional groups present on the sorbent surface. Noble metals occur as negative chloride complexes, namely [AuCl_4_]^−^, [PdCl_4_]^2−^, [PtCl_6_]^2−^, and [RhCl_6_]^3−^, in chloride solutions. The influence of hydrochloric acid concentration on noble metal ions’ sorption was investigated. The results presented in Figure 3 indicate that the concentration of hydrochloric acid affects sorption chloride complexes [AuCl_4_]^−^, [PdCl_4_]^2−^, and [PtCl_6_]^2−^. In the 0.1 M HCl solution, platinum(IV), gold(III), and palladium(II) ions were sorbed effectively. The values of R% were 99.2% for Au(III), 99.3% for Pd(II), and 97.5% for Pt(IV). In the 1 M HCl solution, the values of R% were slightly smaller than in 0.1 M HCl: 95.77%—Au(III), 98.18%—Pd(II), and 93.16%—Pt(IV). An increase in the acid concentration to 3 M HCl caused greater changes in R%. The value of R% decreased the most for Pd ions (79.0%), while concentrations for Au(III) and Pt(IV) ions were 94.3% and 90.7%, respectively. With the 6M HCl concentration, the values of R% were still high for Au(III) (93.4%) and Pt(IV) (86.8%). The largest effect of acid concentration was observed for palladium(II) ions. Its R% was only 20.9%. This indicates that chloride ions compete more strongly with functional groups than palladium(II) ions. A similar effect was observed for ion exchangers with functional quaternary ammonium groups—Purolite A-850 and Amberlite IRA-958—for palladium(II) ions [24].

Four kinetic models, namely pseudo-first-order, pseudo-second-order, Elovich, and intra-particle models, were used to analyze gold(III), platinum(IV), and palladium(II) sorption on the impregnated sorbent Nitrolite–Aliquat 336. The linear form of these models can be described using the following equations:(2)$$\ln\left(q_{e}-q_{t}\right)=lnq_{e}-k_{1}\times t\;(\mathrm{PFO})$$
(3)$$\frac{t}{q_{t}}=\frac{1}{k_{2\cdot }q_{e}^{2}}+\frac{1}{q_{e}}\times t\;(\mathrm{PSO}1)$$
(4)$$q_{e}=\frac{1}{\beta }\times ln\left(\alpha \times \beta \right)+\frac{1}{\beta }\times lnt\;(\mathrm{Elovich})$$
(5)$$q_{t}=k_{ip}\times t^{0.5}+C\;(\mathrm{Intra}\text{-}\mathrm{particle}\;\mathrm{diffusion})$$

On the basis of the above equations, the values of kinetic parameters were calculated. The kinetic parameters are presented in Table 1.

For the investigated noble metal ions, small values of the determination coefficient were obtained based on the intra-particle diffusion model. This indicates that the sorption of platinum(IV), palladium(II), and gold(III) chloride complexes is not limited by intra-particle diffusion. The values of coefficients of determination for the Elovich model are higher than for PFO. The best values of coefficients of determination were obtained for the pseudo-second model. This model best described the kinetics of platinum(IV), palladium(II), and gold(III) ions’ sorption on the impregnated sorbent Nitrolite–Aliquat 336. A similar PSO model correlated the data well during Pt(IV) ions’ sorption on SIRs (XAD7-DB30C10) [25].

### 2.3. Isotherms of Noble Metal Ion Sorption on the Impregnated Sorbent Nitrolite–Aliquat 336

It is very important to determine the sorption capacity of platinum(IV), palladium(II), and gold(III) ions on the impregnated sorbent Nitrolite–Aliquat 336. Sorption capacity allowed for a comparison of the sorption properties of Nitrolite–Aliquat 336 with those of other sorbents. In addition, differences were observed in the sorption capacity for different ions. The isotherms of Au(III), Pd(II), and Pt(IV) ions’ sorption in 0.1 M HCl on the impregnated sorbent Nitrolite–Aliquat 336 in 0.1 M HCl are presented in Figure 4. The isotherms of platinum(IV), palladium(II), and gold(III) sorption were determined for single ions. As shown in Figure 4, the maximum sorption capacity was obtained for gold(III) ions. The sorption capacity for gold(III) ions was estimated to be 73.43 mg/g. The sorption capacities for palladium(II) and platinum(IV) ions were lower than for gold(III) ions. The values of sorption capacity for palladium(II) and platinum(IV) ions were 47.63 mg/g and 51.39 mg/g, respectively. These high values of sorption capacity were obtained because the sorbent Nitrolite did not sorb platinum(IV), palladium(II), and gold(III) ions before impregnation.

Two Langmuir and Freundlich isotherm models were used for the determination of constants for Au(III), Pd(II), and Pt(IV). The parameters were calculated from the following equations:(6)$$\frac{C_{e}}{q_{e}}=\frac{1}{b\cdot Q_{o}}+\frac{C_{e}}{Q_{o}}\;(\mathrm{Langmuir})$$
where *q_e_* is the concentration of noble metal ions in the sorbent (mg/g); *C_e_* is the equilibrium concentration (mg/L); *b* is the Langmuir isotherm constant (L/mg); and *Q*_0_ is the maximum monolayer coverage capacity (mg/g).
(7)$${logq}_{e}={logK}_{F}+\frac{1}{n}logC_{e}\;(\mathrm{Freundlich})$$
where *q_e_* is the concentration of noble metal ions in the sorbent (mg/g); *K_F_* is the characteristic constant related to the adsorption capacity (L/g); *n* is the adsorption intensity; and *C_e_* is the equilibrium concentration (mg/L).

The calculated parameters of the isotherm models for the Au(III), Pd(II), and Pt(IV) sorption on the impregnated Nitrolite–Aliquat 336 are presented in Table 2.

The smallest values of coefficients of determination were obtained for the Freundlich model. This indicates that the sorption of Au(III), Pd(II), and Pt(IV) on the impregnated Nitrolite–Aliquat 336 does not comply with the Freundlich model.

The highest values of coefficients of determination were obtained for the Langmuir model. This isotherm model best described platinum(IV), palladium(II), and gold(III) ions’ sorption on the impregnated Nitrolite–Aliquat 336. The Langmuir model assumes the formation of a monolayer and allows for the determination of sorption capacity. The sorption capacity *Q*_0_ calculated from the Langmuir model is close to the value found in Figure 4. For example, the capacity calculated for gold(III) ions was 73.49 mg/g, while the determined value was 73.43 mg/g. The sorption capacity for Pt(IV) (51.59 mg/g) was higher than for SIR (XAD7-DB30C10), which was 15.03 mg/g as reported in the literature [25]. Mesoporous silica impregnated using (3-(3-(methoxycarbonyl)benzylidene)hydrazinyl)benzoic acid (MBHB) ligand exhibited a larger sorption capacity for palladium(II) ions (184.5 mg/g) than Nitrolite–Aliquat 336 [26]. Moreover, a high sorption capacity of palladium(II) ions (213.67 mg/g) was obtained for a nanoconjugate adsorbent (NCA) [27]. The conjugate adsorbent achieved a high sorption capacity for gold(III) ions, 183.42 mg/g [28]. The maximal capacity of an amine-based functionalized mesoporous silica adsorbent was 94.92 mg Pd/g in a nitrate(V) solution [29]. Palladium ions were effectively sorbed by silica-based IsoBu-BTP/SiO_2_-P adsorbent, obtaining a sorption capacity of 0.71 mmol/g [30]. As can be seen, the impregnated Nitrolite–Aliquat 336 acquired smaller values of sorption capacity than the mentioned sorbents, but its advantage is being a simple method of preparation.

### 2.4. Mechanism of Noble Metal Ion Sorption

The impregnation process allows for the incorporation of quaternary ammonium groups from Aliquat 336 into the Nitrolite sorbent. The sorption process occurs according to the ion exchange mechanism. Ion exchange can be presented using the following equations:[Aliquat 336^+^Cl^−^] + [AuCl_4_]^−^ ⇄ [Aliquat 336^+^][AuCl_4_]^−^ + Cl^−^
2[Aliquat 336^+^Cl^−^] + [PdCl_4_]^2−^ ⇄ [Aliquat 336^+^]_2_[PdCl_4_]^2−^ + 2Cl^−^
2[Aliquat 336^+^Cl^−^] + [PtCl_6_]^2−^ ⇄ [Aliquat 336^+^]_2_[PtCl_6_]^2−^ + 2Cl^−^

The ion exchange mechanism is consistent with the kinetics results. An increase in the acid concentration reduced the sorption capacity, and the reaction equilibrium shifted to the right.

### 2.5. Desorption

Desorption is very important as a sorbent can be used several times. Our research results reveal that the anionic complexes [AuCl_4_]^−^, [PdCl_4_]^2−^, and [PtCl_6_]^2−^ are sorbed by positive functional groups such as quaternary ammonium. Nitrogen has a constant positive charge, and in order to desorb noble metal ions, its charge must be changed to positive. Thiourea allows for the exchange of chloride ligands for thiourea ligands. Therefore, it was used as the desorption solution: 1 M thiourea in 1 M HCl.

As can be seen in Figure 5 noble metal ions were desorbed effectively. Palladium(II) ions were desorbed the best, D%—97.3%. Platinum(IV) (95,4%) and gold(III) (94.1%) ions were equally well desorbed. Similar to our investigations, thiourea in HCl solution was used for the desorption of platinum(IV) and palladium(II) from di-hexylamine solvent-impregnated resin DHA-SIR and di-hexyl sulfide solvent-impregnated resin DHS-SIR, respectively [31].

### 2.6. Recovery of Noble Metals from the Exhausted Catalytic Converter

Exhausted catalytic converters are a valuable source of precious metals [32]. After leaching, a solution contains three platinum metals: platinum(IV), palladium(II), and rhodium(III). The concentrations of these ions were 26.45 mg/L for Pt(IV), 38.85 mg/L for Pd(II), and 7.25 mg/L for Rh(III). As sorbent tests proved, it removes Pt(VI) and Pd(II) ions from the mixture, while Rh(III) ions are not sorbed.

As shown in Figure 6, the impregnated Nitrolite–Aliquat 336 is selective and removes ions of Pt(IV) (97.94%) and Pd(II) (94.3%) from the mixture after leaching. The impregnated sorbent is not suitable for the sorption of rhodium(III) ions. Our future research will focus on obtaining a selective sorbent for rhodium(III) ions.

## 3. Materials and Methods

### 3.1. Materials

The natural sorbent Nitrolite is an aluminosilicate with the following formula: (K,Na,1/2Ca)_2_‧Al_2_O_3_‧10SiO_2_‧H_2_O. The total capacity of Nitrolite is 0.7 eq/L (NH_4_^+^ ions), and the bed size is in the range of 0.63–1.4 mm. Nitrolite was washed prior to experiments in order to remove surface dust.

Commercial-grade extractant Aliquat 336 was used for the impregnation of Nitrolite. The chemical structure of Aliquat 336 is shown in Figure 7. Aliquat 336 is methyl-trioctylammonium chloride, F.W. 404.17 (d = 0.884 g/mL), and it was supplied by Aldrich, Steinheim, Germany.

Noble metal solutions were prepared from H_2_PtCl_6_ 30% (POCH Gliwice, Gliwice, Poland), HAuCl_4_ 30% (POCH Gliwice, Poland), and PdCl_2_ (POCH Gliwice, Poland). Standard solutions for the ICP-OES analysis were diluted from gold(III) (1000 mg/L), palladium(II) (1000 mg/L), rhodium(III) (1000 mg/L), and platinum(IV) (1000 mg/L) reference solution, which was produced by ROMIL (Cambridge, UK). The water used for the investigations was HPLC-grade, and it was obtained from the Polwater DL2-150 system, (Kraków, Poland). Its conductivity was 0.05 µ S/cm.

### 3.2. Instrumentation

The concentration of noble metal ions was determined via ICP-OES (inductively coupled plasma–optical emission spectrometry) method using a Varian 720ES spectrometer. The values of ICP-OES operating parameters are presented in Table 3.

The impregnated sorbent samples were shaken using a laboratory shaker Elpin+ type 358, Lubawa, Poland, at an amplitude of 8 and a speed of 160 c.p.m.

An FTIR-ATR (Fourier transform infrared spectroscopy–attenuated total reflectance) spectrometer Agilent Technologies Cary 630 was used for sample investigations.

FTIR-ATR spectra were recorded for Nitrolite, Aliquat 336, and Nitrolite after its impregnation by Aliquat 336.

A conductivity value was measured using an electrode (type PS-2Z) connected to the Elmetron CP-401 multifunction meter.

Scanning electron microscopy (SEM) measurements were conducted by means of a Quanta 3D FEG scanning electron microscope produced by FEI Company (Oregon, OH, USA).

### 3.3. Methods

#### 3.3.1. Nitrolite Impregnation Procedure

Briefly, 1 g of Aliquat 336 was heated to 333 K. A 2 g sample of Nitrolite was mixed with 1 g of hot Aliquat 336 (ratio of Aliquat 336 g:Nitrolite g—1:2) and stirred for 2 h. The mixture of the impregnated sorbent and solution was washed with distilled water to remove the impregnation solution and then dried at 298 K. The sorbent prepared in this way was used in the research.

#### 3.3.2. Determination of Aliquat 336 Concentration in Nitrolite

Aliquat 336 concentration in Nitrolite was determined via the titration of 0.1957 g Aliquat 336 in an acetone–water medium (12.5 mL acetone + 12.5 mL water) using a 0.1 M AgNO_3_ solution. During titration, the conductivity of the solution was measured. A 0.4069 g sample of Nitrolite impregnated with Aliquat 336 in the acetone–water medium (12.5 mL acetone + 12.5 mL water) was titrated with a 0.1 M AgNO_3_ solution. The titration endpoint was determined from a graph plotting conductivity vs. titrant volume.

#### 3.3.3. Kinetic Studies

First, 0.2 g of Nitrolite–Aliquat 336 was equilibrated with 50 mL of the mixture comprising gold(III) (10 mg/L), palladium(II) (10 mg/L), platinum(IV) (10 mg/L), and rhodium(III) (10 mg/L) at HCl concentrations of 0.1 M, 1 M, 3 M, and 6 M, in a shaker bath at 293 K. The samples were shaken in the contact time range of 1–1440 min.

#### 3.3.4. Isotherm Studies

Briefly, 0.1 g of Nitrolite–Aliquat 336 was equilibrated with 25 mL of individual gold(III), palladium(II), and platinum(IV) ions in the initial concentration range of 10–1000 mg/L, at the HCl concentration of 0.1 M, in a shaker bath at 293 K. The samples were shaken under a contact time of 1440 min.

#### 3.3.5. Catalytic Converter Digestion and Sorption Platinum Metals

An exhausted catalytic converter was digested in hydrochloric acid and peroxide solution. To this end, 20 g of the catalytic converter was crushed and milled. The milled catalytic converter was contacted with 100 mL of water and 100 mL of 36% HCl. The catalyst was mixed with the solution in a flask, and hydrogen peroxide (100 mL 30% H_2_O_2_) was added in portions. The contents of the flask were maintained at 333 K for 2 h. After cooling, the solution was filtered off from the catalyst residue. It was analyzed via ICP-OES to determine concentrations of precious metals. Before sorption, the solution was diluted 10 times.

#### 3.3.6. Desorption Studies

For desorption studies, 0.1 g of sorbent was loaded with gold(III), palladium(II), and platinum(IV) using 10 mL of the 50 mg/L solution in 0.1 M HCl, and the agitation period of 24 h was applied. In addition, 1 M thiourea in 1 M HCl solution was used as a desorption agent. The solution was analyzed using ICP-OES to determine the concentration of precious metals.

## 4. Conclusions

This research involves an investigation of the possibility of preparing a cheap impregnated sorbent Nitrolite–Aliquat 336 for the sorption of noble metal ions such as gold(III), palladium(II), and platinum(IV). The proposed warm impregnation procedure makes it possible to obtain an impregnated sorbent without the use of organic solvents. This method of impregnation enabled the preparation of an impregnated sorbent with a large content of the impregnating agent, i.e., 0.25 g of Aliquat 336/gNitrolite. In the 0.1 M HCl solution, platinum(IV), gold(III), and palladium(II) ions were sorbed effectively, and the values of R% were 99.2% for Au(III), 99.3% for Pd(II), and 97.5% for Pt(IV). An increase in the HCl concentration caused a decrease in the value of R%. The maximum sorption capacity was estimated to be 73.43 mg/g for the sorption of gold(III) ions, 51.39 mg/g for platinum(IV) ions, and 47.63 mg/g for palladium(II) ions from the 0.1 M HCl model solution. The Langmuir isotherm and the PSO kinetic models described the sorption of noble metals well. The prepared impregnated sorbent enabled the selective recovery of palladium(II) and platinum(IV) ions from the solution after leaching of the spent catalytic converter. In the experiment using the real leaching solution, the values of R% were high for Pt(IV) (97.94%) and Pd(II) (94.3%). This sorbent is not suitable for rhodium(III) ions’ sorption. The test results confirmed that Au(III), Pt(IV), and Pd(II) ions can be desorbed from impregnated Nitrolite–Aliquat 336 using 1 M thiourea in the 1 M HCl solution. The values of D% were 97.3%for Pd(II), 95.4% for Pt(IV), and 94.1% for Au(III).

## Figures and Tables

**Figure 1 molecules-28-06040-f001:**
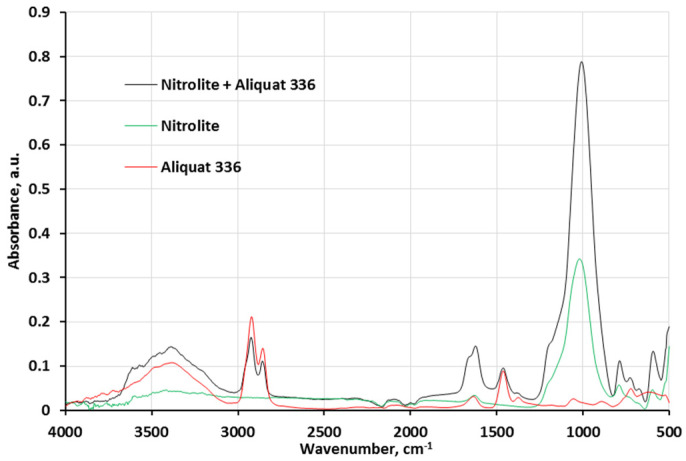
ATR-FTIR spectra for Aliquat 336, Nitrolite and Nitrolite–Aliquat 336.

**Figure 2 molecules-28-06040-f002:**
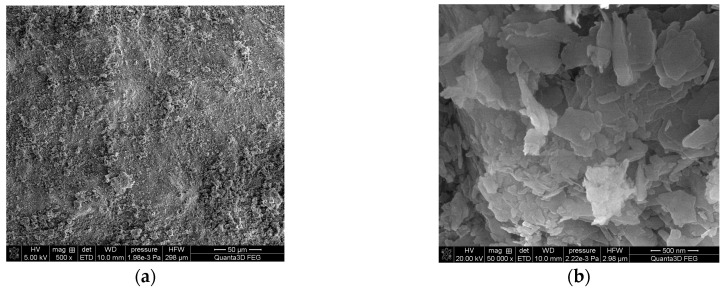
SEM images: (**a**) Nitrolite mag. 500×; (**b**) Nitrolite mag. 50,000×; (**c**) Nitrolite–Aliquat 336 mag. 500×; (**d**) Nitrolite–Aliquat 336 mag. 50,000×.

**Figure 3 molecules-28-06040-f003:**
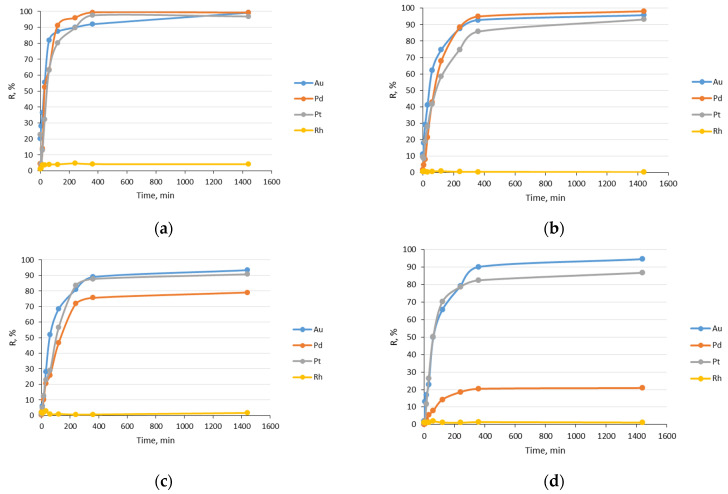
Influence of contact time on %R gold(III), platinum(IV), palladium(II), and rhodium(III) ions’ sorption from hydrochloric acid: (**a**) 0.1 M; (**b**) 1 M; (**c**) 3 M; (**d**) 6 M.

**Figure 4 molecules-28-06040-f004:**
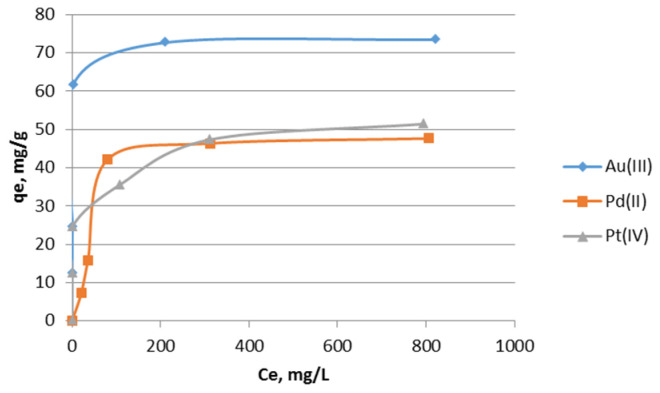
Isotherms of Au(III), Pd(II), and Pt(IV) ions’ sorption in 0.1 M HCl on the impregnated sorbent Nitrolite–Aliquat 336 in 0.1 M HCl; qe is the concentration of noble metal ions in the sorbent (mg/g); Ce is the equilibrium noble metal ion concentration (mg/L).

**Figure 5 molecules-28-06040-f005:**
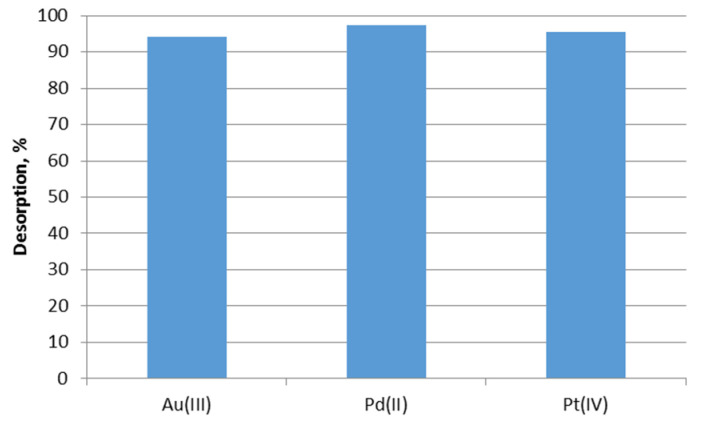
Desorption of Au(III), Pd(II), and Pt(IV) ions from the impregnated sorbent Nitrolite–Aliquat 336 using 1 M thiourea in 1 M HCl.

**Figure 6 molecules-28-06040-f006:**
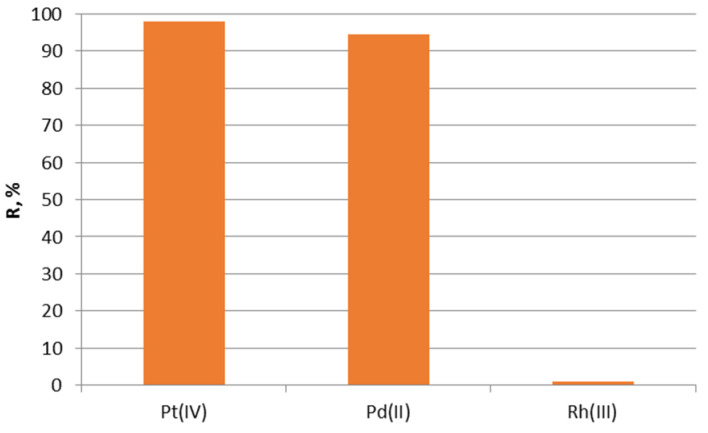
Sorption of Rh(III), Pd(II), and Pt(IV) ions from leaching solution on the impregnated sorbent Nitrolite–Aliquat 336.

**Figure 7 molecules-28-06040-f007:**
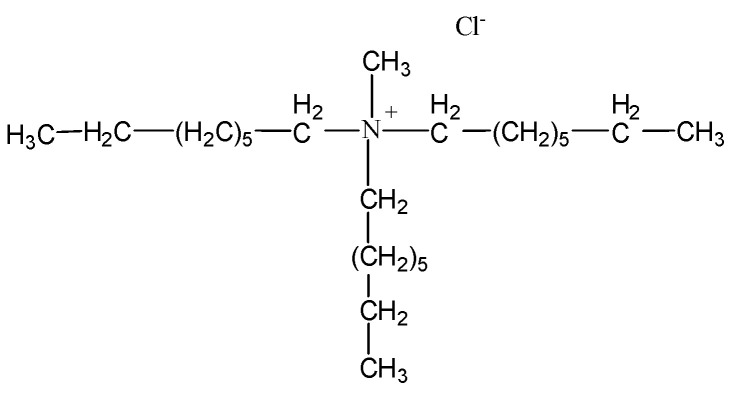
Structure of Aliquat 336.

**Table 1 molecules-28-06040-t001:** Kinetic parameters for the Au(III), Pd(II), and Pt(IV) ions’ sorption on the impregnated sorbent Nitrolite–Aliquat 336.

		PFO			PSO 1		
Metal Ions	HCl (M)	k_1_ (1/min)	q_e_ (mg/g)	R^2^	k_2_ (g/mg‧min)	q_e_ (mg/g)	R^2^
Au(III)	0.1	0.0033	0.9528	0.8765	0.0207	2.5054	0.9996
Pd(II)	0.1	0.0039	0.5295	0.5750	0.0064	2.6203	0.9819
Pt(IV)	0.1	0.0023	1.0949	0.5442	0.0079	2.5284	0.9938
Au(III)	1	0.0024	1.2661	0.7178	0.0144	2.4434	0.9994
Pd(II)	1	0.0035	2.4076	0.8363	0.0040	2.6495	0.9909
Pt(IV)	1	0.0036	2.8527	0.9548	0.0070	2.4216	0.9977
Au(III)	3	0.0033	2.0438	0.8742	0.0063	2.4534	0.9978
Pd(II)	3	0.0030	1.6086	0.8062	0.0021	2.3283	0.9473
Pt(IV)	3	0.0029	1.9874	0.7645	0.0042	2.4534	0.9941
Au(III)	6	0.0042	2.6528	0.9452	0.0068	2.4704	0.9981
Pd(II)	6	0.0030	0.0636	0.7531	0.0123	0.5878	0.9615
Pt(IV)	6	0.0028	1.3925	0.7834	0.0041	2.3656	0.9699
		Elovich			Intra-Particle Diffusion		
Metal Ions	HCl (M)	α (g/mg‧min)	β (mg/g)	R^2^	K (mg/g‧min^0.5^)	C (mg/g)	R^2^
Au(III)	0.1	1.0062	3.0774	0.9076	0.0515	1.0461	0.6029
Pd(II)	0.1	0.3138	2.3182	0.8721	0.0687	0.6597	0.5852
Pt(IV)	0.1	0.3749	2.6850	0.7980	0.0646	0.6428	0.6352
Au(III)	1	0.4582	2.8257	0.9348	0.0590	0.7419	0.6866
Pd(II)	1	0.1960	2.3831	0.8754	0.0744	0.3298	0.7270
Pt(IV)	1	0.2605	2.8944	0.9041	0.0630	0.4356	0.7960
Au(III)	3	0.2276	2.5724	0.9099	0.0668	0.4287	0.7108
Pd(II)	3	0.1548	3.0073	0.8785	0.0600	0.2327	0.7554
Pt(IV)	3	0.1808	2.6212	0.8713	0.0686	0.2877	0.7454
Au(III)	6	0.2441	2.6497	0.9133	0.0662	0.4441	0.7423
Pd(II)	6	0.0540	11.167	0.8911	0.0158	0.0710	0.7340
Pt(IV)	6	0.2104	2.6606	0.8987	0.0631	0.4088	0.6704

**Table 2 molecules-28-06040-t002:** The Langmuir and Freundlich isotherm models’ constants and coefficients of determination for Au(III), Pd(II), and Pt(IV) ions’ sorption on the impregnated sorbent Nitrolite–Aliquat 336.

Metal Ions	Langmuir				Freundlich		
	Q_0_	b	R_L_	R^2^	K_f_	n	R^2^
Au(III)	73.49	0.9229	0.0021	0.9999	26.25	5.40	0.6886
Pd(II)	51.27	0.0187	0.0966	0.9784	2.75	2.11	0.7285
Pt(IV)	51.59	0.0735	0.0264	0.9957	19.76	6.92	0.8890

**Table 3 molecules-28-06040-t003:** ICP-OES parameters.

Parameter	Value
Power (kW)	1
Plasma flow (L/min)	15
Auxiliary flow (L/min)	1.5
Nebulizer flow (L/min)	0.75
Replicate read time (s)	1
Instrument stabilization delay (s)	15
Sample uptake delay (s)	15
Pump rate (r.p.m)	15
Rinse time (s)	15
Au (nm)	242.794
Pd (nm)	340.458
Pt (nm)	214.424
Rh (nm)	343.488

## Data Availability

The data are included in the article and available upon request.

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
