# Peer review of "A New Impregnated Adsorbent for Noble Metal Ion Sorption"

_molecules, 2023, doi:10.3390/molecules28166040_

Round 1

Reviewer 1 Report

file

Minor editing of English language required.

Author Response

Reply to Reviewer 1 Comments

Comments and Suggestions for Authors

obtained a new Nitrolite impregnated by Aliquat 336. The FTIR-ATR analysis was used to confirm the presence of Aliquat 336 in the Nitrolite. The sorbent binds platinum(IV), palladium(II), and gold(III) ions from the model chloride solutions, but does not bind rhodium(III) ions. In the next experiment, Authors leached an exhausted catalytic converter to transfer to solution precious metal ions (Pt(IV), Pd(II), and Rh(III)). The desorption process was also carried out.

The novelty of the work is quite high and the experiments and results are clearly described.

However, some details need explanation or correction. Suggestions and comments on the text

are provided below.

Thank you very much for the kind revision. All comments were included in the revised paper. It allowed to improve the paper significantly.

  1. Abstract Authors should clearly distinguish, which tests were carried out on model solutions and which on real ones. Especially, because the real solution did not contain gold(III) ions and was not similar to the model solution.

Thank you for this comment. Abstract has been improved.

  1. Introduction, lines 47-49 is incomprehensible.

This statement was changed.

Next to extractants containing S, P-donor atoms, amines also can be used for sorption and separation of noble metals.

  1. Results, lines 113-114 What Authors had in mind with water

molecules contained in Aliquat 336?

The sorbent Nitorlite-Aliquat 336 can contain moisture, and this band is visible in the ATR-FTIR spectra.

  1. Results, line 132 Grammar mistake „the a”

Thank you for this remark.  The grammar mistake has been improved.

“from the  gold(III), palladium(II), platinum(IV) and rhodium(III) mixture”

  1. Figure 2, description After „word” please add „sorption”

The word sorption was added.

Figure 2. Influence of contact time on the %R gold(III), platinum(IV), palladium(II) and rhodium(III) ions sorption from hydrochloric acid: (a) 0.1M; (b) 1M; (c) 3M; (d) 6M.

  1. All manuscript Please number all equations.

Thank you for this remark.  The equations have been numbered.

  1. Table 1 The description of Table 1 is mistaken.

Thank you for this remark.  The description of Table 1 has been improved.

Table 1. Kinetic parameters for the Au(III), Pd(II), Pt(IV) ions sorption on the impregnated sorbent Nitrolite-Aliquat 336.

  1. Figure 3, graph Authors should provide an explanation of the abbreviation Ce. The explanation appears on the 7 page.

The explanation has been added.

(qe is the amount of noble metal ion in the sorbent (mg/g); Ce is the equilibrium noble metal ion concentration (mg/L).

  1. Results, line 266 Authors should provide an explanation of the abbreviations DHA-SIR and DHS-SIR.

The explanation has been added.

the di-hexylamine solvent impregnated resin DHA-SIR and  the di-hexyl sulfide solvent impregnated resin  DHS-SIR

  1. Methods, Paragraph 3.3.4. Isotherm studies What Authors had in mind with

concentration of metal ions 10-1000 mg/l? In the results section, there is no description of

the research with different concentrations of metal ions, only with changing HCl concentration.

In order to make the sorption isotherms, the initial concentrations of metal ions were changed in the range of 10-1000 mg/l at the constant 0.1M HCl concentration. On this basis, the sorption isotherms of individual ions were plotted (Fig. 3).

  1. Conclusions Please extend the conclusion section, now it is very general.

The Conclusion section has been extended.

  1. References Some of the reference positions are old (2010

and older), please change these positions on

the more actual

The oldest literature has been changed.

All your comments were valuable and allowed us to improve the quality of our manuscript.

Reviewer 2 Report

The manuscript describes the gold, platinum, palladium, and rhodium from e-waste using impregnated adsorbent which is a good topic and falls in the topic of the journal, however, there are some issues to be addressed. The comments are listed below:

1. The English of the text should be checked

2. The authors must be included new, relevant, and more information about other adsorbent materials. Diverse studies are growing attention for diverse uses for metal recovery as reported by the Awual group according to ScienceDirect. The authors need to indicate such points for a broad range of readers. Moreover, the authors need to cite high-impact articles to make the manuscript high-level. The following specific articles may take be noted in the revision stage of https://doi.org/10.1016/j.jiec.2014.02.053; https://doi.org/10.1016/j.cej.2014.08.028; https://doi.org/10.1016/j.jiec.2013.12.040

3. Comparison between the obtained results and measured in this study with other reported studies should be done and included for more clarity (indicate values not just the number of references).

4. A schematic mechanism describing the adsorption process must be indicated and included (reactions, interactions, etc.)

5. Correct the References using the guide of the Journal. More Conclusions must be included with the best results, and values obtained.

The English language needs to check carefully in the revision stage because of many careless mistakes in many positions.

Author Response

Reply to Reviewer 2 Comments

Thank you very much for the kind revision. All comments were included in the revised version of the paper. It allowed to improve the paper significantly.

  1. The English of the text should be checked

The manuscript has been corrected by the English specialist.

  1. The authors must be included new, relevant, and more information about other adsorbent materials. Diverse studies are growing attention for diverse uses for metal recovery as reported by the Awual group according to ScienceDirect. The authors need to indicate such points for a broad range of readers. Moreover, the authors need to cite high-impact articles to make the manuscript high-level. The following specific articles may take be noted in the revision stage of

https://doi.org/10.1016/j.jiec.2014.02.053;

https://doi.org/10.1016/j.cej.2014.08.028;

https://doi.org/10.1016/j.jiec.2013.12.040

These articles have been cited.

  1. Comparison between the obtained results and measured in this study with other reported studies should be done and included for more clarity (indicate values not just the number of references).

The comparison of sorption capacities is done.

  1. A schematic mechanism describing the adsorption process must be indicated and included (reactions, interactions, etc.)

The mechanism of sorption of noble metals is presented by reactions:

 [Aliquat 336+Cl-] + [AuCl4]- ⇄ [Aliquat 336+][AuCl4]- + Cl-

2[Aliquat 336+Cl-] + [PdCl4]2- ⇄ [Aliquat 336+ ]2[PdCl4]2- + 2Cl-

2[Aliquat 336+Cl-]+ [PtCl6]2- ⇄ [Aliquat 336+ ]2[PtCl6]2- + 2Cl-

This is the ion exchange mechanism of noble metal ions sorption.

  1. Correct the References using the guide of the Journal. More Conclusions must be included with the best results, and values obtained.

The references have been corrected. The Conclusion section has been extended.

All your comments were valuable and allowed us to improve the quality of our manuscript.

Reviewer 3 Report

1. Justify the chosen ratios of the sorbent and liquid phase when studying the kinetics and isotherms.

2. In this work, it is said that Dubinin-Radushkevich Isotherm is used to distinguish between physical and chemical adsorption by measuring the E value. However, other researchers believe that this isotherm does not provide any information about the sorption mechanism [1] since it ignored the influence of the solvent, especially solution pH on the chemical species of the solutes, and surface charge and functional groups dissociation of the sorbents in a solid/solution sorption system. Thus, the Dubinin-Radushkevich isotherm model may not accurately provide the mean free energy to distinguish physical or chemical sorption in a solid/solution sorption system [1].

According to an article [1], there is an obvious misconception regarding the equation Polanyi potential, presented in this article, which must be indicated. The magnitude of the sorption Polanyi potential ε depended on the unit of the equilibrium concentration Ce. In most sorption studies, either mol L−1 or mg L−1 was used as the unit of Ce. The sorption potential obtained if mol L−1 was used would be different from that obtained if mg L−1 was used. Thus, it was incorrect to calculate the mean free energy E using the parameter β obtained by equation (8), as the corresponding mean free energy may give a wrong explanation concerning sorption mechanisms [1].

[1] Hu, Q.; Zhang, Zh. Application of Dubinin-Radushkevich isotherm model at the solid/solution interface: a theoretical analysis. J. Mol. Liq. 2019, 277, 646–648. https://doi.org/10.1016/j.molliq.2019.01.005

3. The conclusions are formulated incorrectly. In the conclusions, it is necessary to briefly give specific results obtained and the patterns identified.

Author Response

Reply to Reviewer 3 Comments

Thank you very much for the kind revision. All comments were included in the revised version of the paper. It allowed to improve the paper significantly.

  1. Justify the chosen ratios of the sorbent and liquid phase when studying the kinetics and isotherms.

The ratio 1:2 allows the entire volume of the Nitrolite to be wetted by Aliquat 336. The use of a larger volume of Aliquat 336 does not improve the impregnation process.

  1. In this work, it is said that Dubinin-Radushkevich Isotherm is used to distinguish between physical and chemical adsorption by measuring the E value. However, other researchers believe that this isotherm does not provide any information about the sorption mechanism [1] since it ignored the influence of the solvent, especially solution pH on the chemical species of the solutes, and surface charge and functional groups dissociation of the sorbents in a solid/solution sorption system. Thus, the Dubinin-Radushkevich isotherm model may not accurately provide the mean free energy to distinguish physical or chemical sorption in a solid/solution sorption system [1].

According to an article [1], there is an obvious misconception regarding the equation Polanyi potential, presented in this article, which must be indicated. The magnitude of the sorption Polanyi potential ε depended on the unit of the equilibrium concentration Ce. In most sorption studies, either mol L−1 or mg L−1 was used as the unit of Ce. The sorption potential obtained if mol L−1 was used would be different from that obtained if mg L−1 was used. Thus, it was incorrect to calculate the mean free energy E using the parameter β obtained by equation (8), as the corresponding mean free energy may give a wrong explanation concerning sorption mechanisms [1].

[1] Hu, Q.; Zhang, Zh. Application of Dubinin-Radushkevich isotherm model at the solid/solution interface: a theoretical analysis. J. Mol. Liq. 2019, 277, 646–648. https://doi.org/10.1016/j.molliq.2019.01.005

Thank you for this remark. This article is very interesting and helpful. We have decided to neglect parameters for the D-R model.

  1. The conclusions are formulated incorrectly. In the conclusions, it is necessary to briefly give specific results obtained and the patterns identified.

The Conclusion section has been extended.

All your comments were valuable and allowed us to improve the quality of our manuscript.

Reviewer 4 Report

The manuscript reports the preparation of new impregnated adsorbent for noble metal ions sorption. There are some interesting results that a high adsorption capacity by this material was achieved. But in my opinion there are still many problems which should be improved before it is considered to be published.

1.      The SEM characterization should be given to confirm morphology.

2.      The title of table 1 is wrong that it should be the kinetics parameters.

3.      The reusability and stability performance of the adsorbent should be evaluated.

4.      Line 325 “ 1 g of Aliquat 336 was heated to 60 0C” the purpose of heating should be explained.

5.      “100cm3” mL is suggested to be used.

6.      Line 133 “from the a gold(III), palladium(II), platinum(IV) and rhodium(III) mixture.” Grammer mistakes that the and a can’t be used together.

7.      To fully improve the quality of this manuscript, please refer to the following article:

(a)    Precise separation and efficient enrichment of palladium from wastewater by amino-functionalized silica adsorbent, Journal of Cleaner Production 396 (2023) 136479

(b)    Separation and recovery of Rh, Ru and Pd from nitrate solution with a silica-based IsoBu-BTP/SiO2-P adsorbent

The languague should be improved.

Author Response

Reply to Reviewer 4 Comments

The manuscript reports the preparation of new impregnated adsorbent for noble metal ions sorption. There are some interesting results that a high adsorption capacity by this material was achieved. But in my opinion there are still many problems which should be improved before it is considered to be published.

  1. The SEM characterization should be given to confirm morphology.

Thank you for this remark. We agree that the SEM images would help characterize the sorbent. The SEM images have been done.

  1. The title of table 1 is wrong that it should be the kinetics parameters.

Thank you for this remark. The title has been improved.

Table 1. Kinetic parameters for the Au(III), Pd(II), Pt(IV) ions sorption on the impregnated sorbent Nitrolite-Aliquat 336.

  1. The reusability and stability performance of the adsorbent should be evaluated.

Thank you for this comment.

The warm impregnation process has been tested by us. This process is repeatable and enables the impregnation of the natural Nitrolite sorbent. Losses of the impregnating agent during the impregnated sorbent operation can take place. This is a disadvantage of the impregnated sorbents. This problem can be solved using the precipitation of water soluble poly(vinyl alcohol) onto the impregnated sorbent with the subsequent crosslinking of poly(vinyl alcohol) layer with vinyl sulphone. We are very sorry but we did not stabilize the impregnated sorbent. We plan to perform and publish this type of research in the future. In this paper, we focused on presenting a new method of warm impregnation.

  1. Line 325 “ 1 g of Aliquat 336 was heated to 60 0C” the purpose of heating should be explained.

The viscosity of Aliquate 336 is 1500mPs/s at 303K and it is difficult to mix with sorbent. When heated to 60 0C, it can be easily mixed, which enables the impregnation process.

  1. “100cm3” mL is suggested to be used.

The unit cm3 is changed into mL throughout the paper.

  1. Line 133 “from the a gold(III), palladium(II), platinum(IV) and rhodium(III) mixture.” Grammer mistakes that the and a can’t be used together.

Thank you for this remark. The grammar mistake has been improved.

“from the gold(III), palladium(II), platinum(IV) and rhodium(III) mixture”

  1. To fully improve the quality of this manuscript, please refer to the following article:

 (a)    Precise separation and efficient enrichment of palladium from wastewater by amino-functionalized silica adsorbent, Journal of Cleaner Production 396 (2023) 136479

 (b)    Separation and recovery of Rh, Ru and Pd from nitrate solution with a silica-based IsoBu-BTP/SiO2-P adsorbent

Thank you for this remark. The listed articles have been cited.

All your comments were valuable and allowed us to improve the quality of our manuscript.

Round 2

Reviewer 2 Report

The authors revised the manuscript based on the remarks to improve the manuscript for the reader’s understanding.

Accept.

Reviewer 4 Report

 The manuscript has been sufficiently improved to be suggested to publication in Molecules.